# EFFECTIVE MODEL PRUNING

## ABSTRACT

We introduce Effective Model Pruning (EMP), a context-agnostic, parameter-free rule addressing a fundamental question about pruning: how many entries to keep. EMP does not prescribe how to score the parameters or prune the models; instead, it supplies a universal adaptive threshold that can be applied to any pruning criterion: weight magnitude, attention score, KAN importance score, or even feature-level signals such as image pixel, and used on structural parts or weights of the models. Given any score vector $s$, EMP maps $s$ to a built-in effective number $N_{eff}$ which is inspired by the Inverse Simpson index of contributors. Retaining the $N_{eff}$ highest scoring entries and zeroing the remainder yields sparse models with performance comparable to the original dense networks across MLPs, CNNs, Transformers/LLMs, and KAN, in our experiments. By leveraging the geometry of the simplex, we derive a tight lower bound on the preserved mass $s_{eff}$ (the sum of retained scores) over the corresponding ordered probability simplex associated with the score vector $s$. We further verify the effectiveness of $N_{eff}$ by pruning the model with a scaled threshold $\beta N_{eff}$ across a variety of criteria and models. Experiments suggest that the default $\beta = 1$ yields a robust threshold for model pruning while $\beta \neq 1$ still serves as an optional adjustment to meet specific sparsity requirements.

## 1 INTRODUCTION

Deep Neural Networks have achieved remarkable results across numerous domains such as computer vision (Krizhevsky et al., 2012; He et al., 2016; Dosovitskiy et al., 2021), natural language processing (Vaswani et al., 2017; Devlin et al., 2019), robotics (Zitkovich et al., 2024), and generative artificial intelligence (Ho et al., 2020), through the deployment of increasingly large and complex models. While this growth has led to more accurate and generalizable models, it also introduces significant deployment challenges on edge devices due to high demands on computation, memory, and energy. Lack of enough resources is particularly evident when deploying large language models (LLMs) (Touvron et al., 2023a;b; Bai et al., 2023) and other over-parameterized models (Liu et al., 2023) in latency-sensitive or resource-constrained environments.

To address deployment challenges of such over-parameterized models, pruning has emerged as a fundamental and widely studied technique (Frankle & Carbin, 2019; Han et al., 2016; Cheng et al., 2024). Pruning has developed a rich taxonomy, typically categorized along three dimensions: what to prune (unstructured weights (Han et al., 2016), structured filters/channels (Li et al., 2017; Liu et al., 2017), or attention heads (Michel et al., 2019; Voita et al., 2019)), when to prune (before (Lee et al., 2019; Wang et al., 2020), during (Louizos et al., 2018; Evci et al., 2020), or after training (Frantar & Alistarh, 2023; Sun et al., 2023)), and how to score parameters (e.g., by magnitude (Han et al., 2016), sensitivity (LeCun et al., 1990; Hassibi et al., 1993), or data-driven metrics (Molchanov et al., 2017)). Despite extensive research in model pruning, a critical and persistent question remains: given a score vector $s$ derived from a pruning criterion, how many candidates should be retained?

The choice of sparsity budget is sensitive. An overly aggressive budget degrades model performance, while an overly conservative one forfeits potential efficiency gains. Current solutions remain unsatisfactory, as sparsity often relies on expensive iterative pruning procedures (Renda et al., 2020), manual or heuristic per-layer budgets, or hyperparameters that require careful tuning (Gale et al., 2019; Frantar & Alistarh, 2023). Recent work (Zhang et al., 2025) gives a sharp lower and upper bound for the pruning rate with specific change of loss tolerance $\epsilon$.

In this paper, we develop Effective Model Pruning (EMP) as a new method to determine retention directly from the score distribution. EMP is a simple rule that automatically determines the effective number $N_{eff}$ of candidates to retain. For any score vector $s$ given by the criterion, EMP computes its effective number $N_{eff}$, inspired by the participation ratio in statistical physics and the inverse Simpson index in ecology (Mézard & Montanari, 2009; Laakso & Taagepera, 1979). This value $N_{eff}$ intuitively represents the number of truly significant contributors. By keeping the top $N_{eff}$ entries, EMP provides a simple computational criterion for deciding how many highest-scoring contributors to keep, in tandem with a tight theoretical lower bound on the retained mass, derived in Section 4.2.

EMP is a universal rule, agnostic of network architecture and pruning paradigm. It eliminates the need for manual budget scheduling and hyperparameter tuning, providing a versatile, robust and automatic pruning limit criterion. To validate the robustness of EMP, we examine the model's performance across a diverse range of criteria and network structures by pruning the model's entries by $\beta N_{eff}$ across a range of scaling coefficients $\beta$. Empirical results demonstrate that models pruned by EMP consistently achieve competitive performance with their dense counterparts, underscoring its effectiveness and generality.

Our contributions are as follows:

- We develop Effective Model Pruning, a simple rule to convert any score vector $s$ into a principled sparsity threshold $N_{eff}$, supported by a theoretically guaranteed lower bound on the preserved mass $s_{eff}$.
- We deduce a lower bound for the loss change $\epsilon$ between dense model and the sparse model with EMP pruning.
- We demonstrate the effectiveness of EMP across diverse architectures and pruning criteria, suggesting it may be combined with existing criteria to achieve strong performance without additional tuning.

## 2 RELATED WORK

### 2.1 PRUNING CRITERIA

Optimal Brain Damage (LeCun et al., 1990) and Optimal Brain Surgeon (Hassibi et al., 1993) estimate the loss increase caused by removing a parameter through second order approximations, and thereby prioritize removals that minimally perturb the objective. Magnitude based heuristics (Han et al., 2016) emerged as a simple and robust baseline in practice and were integrated into end to end compression pipelines that combine pruning with quantization and entropy coding. Empirical study (Gale et al., 2019) confirmed that magnitude-based criteria remain competitive across architectures when combined with careful scheduling and calibration.

Post-training pruning, which is particularly attractive for LLMs due to the prohibitive cost of retraining from scratch, has recently focused on simple but highly scalable criteria. SparseGPT (Frantar & Alistarh, 2023) performs one shot pruning with local least squares reconstruction to control the induced error in each block and achieves strong perplexity at high sparsity without prolonged fine tuning. Wanda (Sun et al., 2023) introduces an activation-aware magnitude score that multiplies absolute weights by a norm of the corresponding activation statistics, thereby adapting the criterion to the data distribution seen at inference. These approaches retain the practical appeal of magnitude-based rules while injecting task awareness through reconstruction or activation weighting.

### 2.2 WHAT TO PRUNE

Unstructured pruning (Han et al., 2016; Gale et al., 2019) removes individual weights and maximizes flexibility in shaping sparsity patterns, while structured pruning removes entire computational units and thereby preserves dense tensor shapes that map efficiently to commodity accelerators. Representative methods in CNNs target filters (Li et al., 2017; He et al., 2019), channels (Luo et al., 2017), or neurons using criteria based on magnitude, batch normalization scaling factors (Liu et al., 2017), or Taylor approximations of the loss (Molchanov et al., 2017). In Transformer architectures, structured pruning often targets attention heads and intermediate feed forward channels. Empirical

analyses showed that many heads are redundant for downstream tasks and can be excised with limited effect (Michel et al., 2019; Voita et al., 2019), while more recent large language model pipelines integrate structured removal of heads, MLP channels, or even layers with light recovery to obtain compact models amenable to further distillation or continued pretraining (Ma et al., 2023; Xia et al., 2024).

Semi-structured pruning strikes a compromise between irregular flexibility and hardware friendliness by enforcing local patterns such as $N:M$-sparsity within rows or columns, which aligns with sparse tensor core primitives on modern GPUs. Learning and representing such patterns efficiently has been an active area of systems and algorithms research (Zhou et al., 2021; Castro et al., 2023). In practice, the choice among unstructured, structured, and semi-structured targets is driven by the deployment stack: when wall clock latency and throughput are paramount, structured or $N:M$-patterns commonly yield more predictable gains (Gale et al., 2019; Cheng et al., 2024).

## 3 PRELIMINARY

In this section, we review the relationship between the sparsity and the model sharpness given by (Zhang et al., 2025, Lemma 3.5), appearing here as Lemma 1.

Let $\hat{y} = f(\theta, x)$ denote a well-trained dense deep neural network with weights $\theta^* \in \mathbb{R}^N$ and empirical loss $L(\theta^*)$. A pruned network derived from the dense network, whose weight is given by $\theta^k = \theta^* \odot M$, where $M$ is a binary mask matrix with $\|M\|_0 = k$ and $\odot$ is entrywise multiplication. Then the pruning ratio $\rho$ is defined as $\rho \triangleq k/N$.

**Lemma 1.** *Given a well-trained neural network $f(\theta^*, x)$, let $\epsilon$ denote the loss difference, $|L(\theta^*) - L(\theta^k)|$, between the dense network and its pruned version, and let $H$ denote the Hessian matrix of the loss function $L$ with respect to the parameter, $\theta$. Then,*

$$\rho \leq 1 - \frac{2\epsilon N}{\left\|\theta^* - \theta^k\right\|_2^2 \operatorname{Tr}(H) + 2\epsilon N},\tag{1}$$

*where $\operatorname{Tr}(H)$ is the trace of the matrix $H$.*

Using Lemma 1, an upper bound for the loss change $\epsilon$ between the dense network and the EMP-pruned model is derived in Section 4.3, since EMP offers a built-in $N_{eff}$-sparse threshold.

## 4 EFFECTIVE MODEL PRUNING

### 4.1 EFFECTIVE POPULATION SIZE AND THE GEOMETRY OF THE SIMPLEX

Fix $N > 1$. Let $s \triangleq (s_1, \ldots, s_N)$ be a vector of scores associated with a pruning object. Define the normalized probability weight vector $\omega$ via

$$\omega_i \triangleq \frac{|s_i|}{\sum_i |s_i|}, \; i = 1, \ldots, N.\tag{2}$$

Then the effective population size $N_{eff} = N_{eff}(\omega)$ is defined as

$$N_{eff} \triangleq \left\lfloor \frac{1}{\sum_i \omega_i^2} \right\rfloor.$$

This section will focus on the geometric interpretation of $N_{eff}$. Consider the standard $(N-1)$-simplex $\Delta$ in the Euclidean space $E = \mathbb{R}^N$ and the affine hyperplane $\Pi$ it spans:

$$\Delta \triangleq \left\{\omega \in \mathbb{R}_{\geq 0}^N : \omega^\top \mathbf{1}_N = 1\right\}, \quad \Pi \triangleq \left\{\omega \in \mathbb{R}^N : \omega^\top \mathbf{1}_N = 1\right\}.\tag{3}$$

Thus, the vector $\omega$ constructed in equation 2 is a point of $\Delta$. Since both $\Delta$ and $N_{eff}$ are invariant under coordinate permutations, for any point $\omega \in \Delta$ and any $\nu \in [N]$, the coordinates can always be permuted so that the first $\nu$ coordinates are the largest $\nu$ weights. More precisely, if $\mathfrak{S}_N$ is the group of permutations on the set $[N] \triangleq \{1, \ldots, N\}$, then

$$\Delta = \bigcup_{\tau \in \mathfrak{S}_N} L_\sigma(\tilde{\Delta}), \quad \tilde{\Delta} \triangleq \{\omega \in \Delta : \omega_1 \geq \omega_2 \geq \cdots \geq \omega_N\},\tag{4}$$

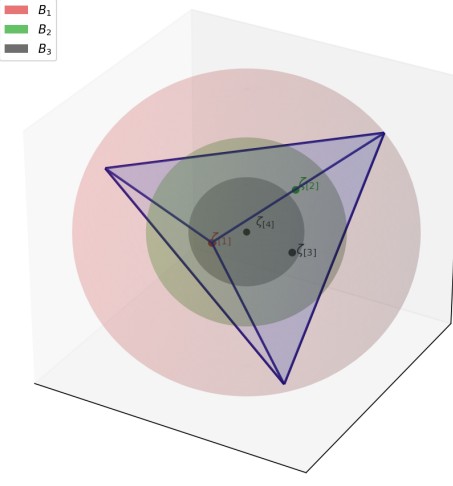

Figure 1: Illustration of the $B_\nu$ balls ($\nu = 1, 2, 3, 4$) and the simplex $\Delta$. Note that ball $B_4$ degenerates to the barycenter $\zeta_{[4]}$.

where $L_\tau : E \to E$ is the linear transformation satisfying $L_\tau(\mathrm{e}_i) = \mathrm{e}_{\tau(i)}$ for all $i \in [N]$. It follows that $L_\tau(\Delta) = \Delta$ and $N_{eff}(\omega) = N_{eff}(L_\tau \omega)$ for all $\tau \in \mathfrak{S}_N$ and all $\omega \in \Delta$. Note also that $L_\sigma(\tilde{\Delta})$ and $L_\tau(\tilde{\Delta})$ are geometric $(N-1)$-dimensional simplices with disjoint interiors whenever $\sigma, \tau \in \mathfrak{S}_N$ and $\sigma \neq \tau$.

The effective mass $s_{eff}$ may then be defined as follows: given $s$, compute $\omega = \omega(s)$ and find $\sigma \in \mathfrak{S}_N$ such that $L_\sigma(\omega) \in \tilde{\Delta}$; then

$$s_{eff} \triangleq \sum_{i=1}^{N_{eff}} \omega_{\sigma(i)}. \tag{5}$$

It follows that $s_{eff} = (L_\tau s)_{eff}$ for all $\tau \in \mathfrak{S}_N$, which makes it sufficient to study $s_{eff}$ restricted to $\tilde{\Delta}$, where one has the simplified formula

$$\omega \in \tilde{\Delta} \implies s_{eff} = \sum_{i=1}^{N_{eff}} \omega_i. \tag{6}$$

By its definition, the effective population size may be characterized as follows. Letting

$$A_\nu \triangleq \left\{ \omega \in \tilde{\Delta} : \nu \leq \|\omega\|^{-2} < \nu + 1 \right\}, \tag{7}$$

one observes that

$$N_{eff}(\omega) = \nu \iff \omega \in A_\nu. \tag{8}$$

Computing a lower bound on $s_{eff}$ in terms of $N_{eff}$ is then tantamount to calculating,

$$\inf_{\omega \in A_\nu} \varphi_\nu(\omega), \text{ where } \varphi_\nu(\omega) \triangleq \sum_{i=1}^{\nu} \omega_i = \nu \left\langle \omega \,\middle|\, \zeta_{[\nu]} \right\rangle, \tag{9}$$

and where

$$\zeta_J \triangleq \frac{1}{|J|} \sum_{i \in J} \mathrm{e}_i, \; J \subseteq [N], \tag{10}$$

denotes the barycenter of the simplex $\mathrm{conv}(\{\mathrm{e}_i : i \in J\})$. The challenge is that this optimization problem is not convex, due to the right-hand side inequality in equation 7. From the identity

$$a, b \in \Delta \implies \left\langle a - \zeta_{[N]} \,\middle|\, b - \zeta_{[N]} \right\rangle = \left\langle a \,\middle|\, b \right\rangle - \frac{1}{N}, \tag{11}$$

it follows that $A_\nu = \tilde{\Delta} \cap (B_\nu - B_{\nu+1})$, where

$$B_\nu \triangleq \left\{ \omega \in \Pi : \left\| \omega - \zeta_{[N]} \right\|^2 \leq \frac{1}{\nu} - \frac{1}{N} \right\}, \tag{12}$$

for $\nu \in [N]$. Thus, $\varphi_\nu$ needs to be minimized over the intersection of $\tilde{\Delta}$ with the spherical shell in $\Pi$ obtained by subtracting the ball $B_{\nu+1}$ from the ball $B_\nu$. Note that $\zeta_{[N]}$ is both the barycenter of $\Delta$ and a vertex of $\tilde{\Delta}$. The vertices of $\tilde{\Delta}$ are precisely all the $\zeta_{[j]}$, $j \in [N]$, with $\zeta_{[1]}, \ldots, \zeta_{[\nu-1]}$ lying outside $B_\nu$, $\zeta_{[\nu]}$ lying on its boundary, and $\zeta_{[\nu+1]}, \ldots, \zeta_{[N]}$ lying in its interior. Each $B_\nu$ is a Euclidean ball in $\Pi$ of radius $r_\nu \triangleq \sqrt{\frac{1}{\nu} - \frac{1}{N}}$ about $\zeta_{[N]}$, and it is tangent at $\zeta_{[\nu]}$ to the $(\nu-1)$-dimensional face of $\tilde{\Delta}$ given by $\mathrm{conv}\big(\zeta_{[\nu]}, \zeta_{[\nu-1]}, \ldots, \zeta_{[1]}\big)$.

One must pay attention to the boundary cases, though. If $\nu = 1$, then $B_\nu$ contains all of $\tilde{\Delta}$ (and hence all of $\Delta$), while $B_{\nu+1} = B_2$ is the ball about $\zeta_{[N]}$ in $\Pi$ "caged" by the edges of $\Delta$. If $\nu = N$, then $B_\nu$ degenerates to a single point, $\zeta_{[N]}$. Finally, for $\nu = N - 1$, $B_\nu$ is the ball about $\zeta_{[N]}$ in $\Pi$, inscribed in $\Delta$, see Figure 1.

## 4.2 Lower Bound on the Effective Mass

With the observations of Section 4.1, a trivial lower bound on $s_{eff}$ is obtained by observing that

$$\inf_{\omega \in A_\nu} \varphi_\nu(\omega) \geq \inf_{\omega \in \tilde{\Delta}} \varphi_\nu(\omega) = \min_{i \in [N]} \nu \left\langle \zeta_{[i]} \,\middle|\, \zeta_{[\nu]} \right\rangle = \nu \left\langle \zeta_{[N]} \,\middle|\, \zeta_{[\nu]} \right\rangle = \frac{\nu}{N}, \tag{13}$$

since expanding the minimization domain to $\tilde{\Delta}$ makes the problem convex. In other words, one always has $s_{eff} \geq \frac{N_{eff}}{N}$.

This paper deduces, and then relies on, a new sharp lower bound as indicated by the following proposition.

**Proposition 1.** *The following bounds hold for $\nu \in \{1, N\}$:*

$$\inf_{\omega \in A_1} \varphi_1(\omega) = \varphi_1(\zeta_{[2]}) = \frac{1}{2}, \quad \inf_{\omega \in A_N} \varphi_N(\omega) = \varphi_N(\zeta_{[N]}) = 1.$$

*Otherwise, if $2 \leq \nu \leq N - 1$, setting the point $p_\nu \in \tilde{\Delta}$ as*

$$p_\nu = \zeta_{[N]} + \frac{r_{\nu+1}}{r_1}(\zeta_{[1]} - \zeta_{[N]}) \in \overline{A_\nu},$$

*the following equality holds:*

$$\inf_{\omega \in A_\nu} \varphi_\nu(\omega) = \varphi_\nu(p_\nu) = \frac{\nu}{N} + \frac{N - \nu}{N}\sqrt{\frac{N - \nu - 1}{(\nu + 1)(N - 1)}}. \tag{14}$$

A proof of Proposition 1 covering the cases $\nu \geq 2$ is presented in Appendix A. Since we are interested in regimes where $N_{eff} = \rho N$, $N \gg 1$, the proof for the $\nu = 1$ case is omitted.

In terms of $s_{eff}$, together with the observations of Section 4.1, Proposition 1 yields the following theorem.

**Theorem 2.** *For all non-zero $s \in \mathbb{R}^N$ with $2 \leq N_{eff} < N$, one has the inequality*

$$1 - s_{eff} \leq \frac{N - N_{eff}}{N}\left(1 - \sqrt{\frac{N - N_{eff} - 1}{(N_{eff} + 1)(N - 1)}}\right) \approx \frac{N - N_{eff}}{N}\left(1 - \sqrt{\frac{N - N_{eff}}{N N_{eff}}}\right). \tag{15}$$

## 4.3 Upper bound of the Performance drop by using EMP

A lower bound on the effective mass is essential for bounding the performance drop in the transition from the dense well-trained network to a pruned network. In this section we study the case where the score vector $s$ coincides with the parameter vector, $\theta$ of the network (in other words, the parameters are scored according to their magnitude). Invoking Lemma 1 with $k = N_{eff}$, one has

$$\rho = \frac{N_{eff}}{N} \leq 1 - \frac{2\epsilon N}{\left\|\theta^* - \theta^k\right\|_2^2 \mathrm{Tr}(H) + 2\epsilon N}. \tag{16}$$

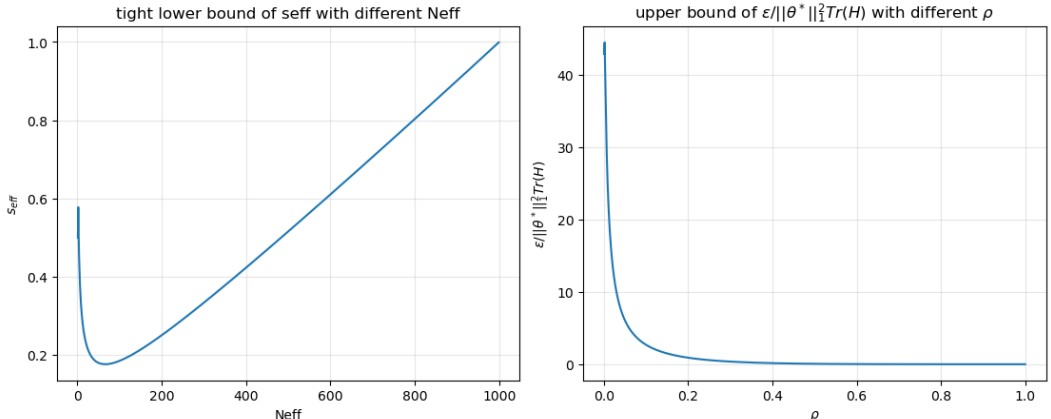

Figure 2: Lower and upper bounds associated with pruning. The left panel illustrates the tight lower bound of the effective mass $s_{eff}$ as a function of $N_{eff}$ for $N = 1000$. The right panel depicts the normalized upper bound of the loss change, $\epsilon/(\|\theta^*\|_1^2 \operatorname{Tr}(H))$, showing its rapid decay as $\rho$ increases.

Rearranging equation 16 yields

$$\epsilon \le \frac{1-\rho}{2N\rho} \operatorname{Tr}(H) \left\|\theta^* - \theta^{N_{eff}}\right\|_2^2, \tag{17}$$

where $\left\|\theta^* - \theta^{N_{eff}}\right\|_2^2$ can be bounded by

$$\begin{aligned}
\left\|\theta^* - \theta^{N_{eff}}\right\|^2 &\le \left\|\|\theta\|_1 (\omega^* - \omega^k)\right\|^2 \\
&\le \left\|\theta^*\right\|_1^2 \left\|(1 - s_{eff}) \mathbf{1}_{[N-N_{eff}]}\right\|^2 \\
&= \left\|\theta^*\right\|_1^2 (1 - s_{eff})^2 (N - N_{eff}).
\end{aligned}$$

Hence, the asymptotic upper bound (as $N \to \infty$) of the loss change $\epsilon$ is

$$\epsilon \lesssim \left\|\theta^*\right\|_1^2 \operatorname{Tr}(H) \frac{(1-\rho)^4}{2\rho} \left(1 - \sqrt{\frac{1-\rho}{N\rho}}\right)^2. \tag{18}$$

The right panel of Figure 2 shows the relationship between $\epsilon/(\|\theta^*\|_1^2 \operatorname{Tr}(H))$ and $N_{eff}$. For $N = 1000$, the value of $\epsilon/(\|\theta^*\|_1^2 \operatorname{Tr}(H))$ is almost equal to 0 if $\rho > 0.2$.

We test the performance of pruning Fully-Connected networks (FCs), AlexNet (Krizhevsky et al., 2012) and VGG16 (Simonyan & Zisserman, 2015) on CIFAR10 (Krizhevsky, 2009), ResNet18 and ResNet50 (He et al., 2016) on CIFAR100, and TinyImageNet (Krizhevsky & Hinton, 2024) with $N_{eff}$ threshold in Section 5.1. As shown in Table 1, test outcomes indicate that the loss change between the dense network and the corresponding EMP pruned network is almost 0 ($\epsilon \le 0.1$).

Note that the upper bound on $\epsilon$ is derived under the weight-magnitude pruning criterion, and this guarantee does not extend to alternative pruning strategies.[1] In contrast, the lower bound on the effective mass $s_{eff}$ can be generalized across different criteria, thereby providing potential upper bounds that quantify performance differences.

## 4.4 EMP ALGORITHM

We provide the pseudo-code of the proposed EMP approach in Algorithm 1. Given a score matrix $S$, the probabilistic vector $\omega$ is derived by normalizing the absolute value $|S|$ with its 1-norm. We

---

[1]Nevertheless, one expects that, for a known and sufficiently smooth scoring function, bounds on first and second derivatives could be used for deriving principles analogous to the one reflected in equation 18.

---

**Algorithm 1** Effective Model Pruning

---

**Require:** Score matrix $S \in \mathbb{R}^N$, coefficient $\beta > 0$
**Ensure:** Binary mask $M \in \{0, 1\}^N$
1: $\omega \leftarrow |S|/\|S\|_1$        ▷ normalize the magnitude of score vector $s$
2: $N_{eff} \leftarrow \lfloor 1/\sum_i^N \omega_i^2 \rfloor$        ▷ get the effective number $N_{eff}$
3: $N_{eff} \leftarrow \text{clip}(\beta N_{eff}, 1, N)$
4: $M \leftarrow \mathbf{0}_N$
5: $\pi \leftarrow \text{argTopK}(|S|, N_{eff})$      ▷ indices of the $N_{eff}$ largest candidates in $|S|$
6: **for** $i \in \pi$ **do**
7:    $M_i \leftarrow 1$
8: **return** $M$

---

Table 1: Loss change between the dense models and the corresponding EMP pruned models.

| Dataset | Model | Dense Loss | Sparsity(%) | EMP Loss | $\epsilon$ |
|---------|-------|------------|-------------|----------|---|
| CIFAR10 | FC5 | 1.2582 | 47.41 | 1.2384 | 0.0198 |
|  | FC12 | 1.5123 | 42.89 | 1.4454 | 0.0669 |
|  | AlexNet | 0.4664 | 62.22 | 0.4286 | 0.0378 |
|  | VGG16 | 0.4234 | 59.47 | 0.3184 | 0.1050 |
| CIFAR100 | ResNet18 | 0.8740 | 56.20 | 0.9287 | 0.0547 |
|  | ResNet50 | 0.8586 | 54.74 | 0.8387 | 0.0199 |
| TinyImagenet | ResNet18 | 2.3028 | 53.37 | 2.2814 | 0.0214 |
|  | ResNet50 | 2.0213 | 48.10 | 1.9853 | 0.0360 |

then compute the effective number $N_{eff}$ and multiply with the coefficient $\beta$, which is an option to meet the specific sparse requirement in practical deployment. The optional coefficient $\beta$ also helps to verify the robustness of $N_{eff}$ by range $\beta \in [0.5, 2.0]$. Since we multiply a potential larger than 1 coefficient, the effective number $N_{eff}$ needs to be constrained within the range of $[1, N]$. We then build a binary mask $M \in \{0, 1\}^N$. Set the $i$-th entry of $M$ to 1 for every $i \in \pi$. The algorithm has a time complexity of $O(N \log N)$.

## 5 EXPERIMENTS

In this section, we show that EMP can be applied to different network architectures, pruning criteria, and pruning objects. Through different values of the coefficient $\beta$, we demonstrate $N_{eff}$ is a robust effective pruning threshold across criterion and architectures. We examine the EMP method across several model types: FCs and CNNs in Section 5.1, Kolmogorov-Arnold Networks (KAN) in Section 5.2, and Large Language Models (LLMs) in Section 5.3. Featurewise pruning results are presented in Section 5.4.

### 5.1 FC MODELS AND CNNS

We first confirm that the $N_{eff}$ threshold yields negligible loss differences $\epsilon$ between the dense model and the EMP-pruned model. Specifically, we evaluate FC5, FC12, AlexNet, and VGG16 on CIFAR-10, as well as ResNet18 and ResNet50 on CIFAR-100 and TinyImageNet, using the $N_{eff}$ threshold in conjunction with the magnitude pruning criterion.

Table 1 demonstrates that the loss difference between the dense network and the EMP pruned network are little ($\epsilon \leq 0.105$) across all tested models. To examine the robustness of $N_{eff}$ as a pruning threshold across different model architectures, EMP is applied on FC2, FC5 and FC12 which are trained on MNIST and Fashion-MNIST. We report the detail training setting in Appendix B.1.

Figure 3 indicates that $\beta = 1$ is the optimal setting across all tested models. For $\beta < 1$, the model will prune more entries than $N_{eff}$, which results model accuracy consistently declines across all configurations. Conversely, for $\beta > 1$, accuracy plateaus, suggesting that any further increase

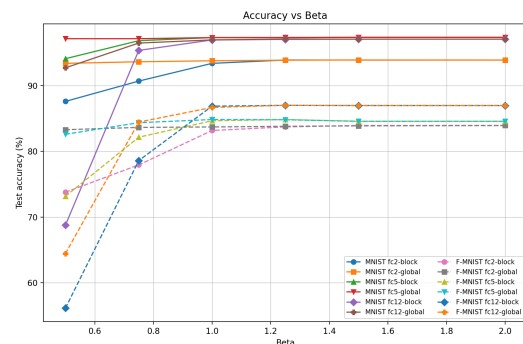

Figure 3: Test Accuracy of EMP-pruned models across different values of $\beta$. We examine 6 discrete values of $\beta = \{0.5, 0.75, 1, 1.25, 1.5, 2\}$ to demonstrate that $N_{eff}$ is a robust pruning threshold across different models and methods and tested on MNIST (solid) and Fashion-MNIST (dashed) datasets.

would unnecessarily reduce model sparsity without yielding performance gains. To fit the specific sparsity requirement for the hardware, $\beta$ can still serve as an optional adjustment.

## 5.2 KAN

From Liu et al. (2025) Section 2.5.1, the incoming score for $i$-th node in $l$-th layer and the outgoing score are defined as

$$I_{l,i} = \max_k(\|\phi_{l-1,i,k}\|_1), \quad O_{l,i} = \max_j(\|\phi_{l+1,j,i}\|_1).$$

In Liu et al. (2025), each node in the network is pruned when both the incoming score and the outgoing score fall below a predefined threshold $\theta$. Instead of this predefined hyperparameter $\theta$, we combine EMP with the criterion $s = \min\{I_{l,i}, O_{l,i}\}$ and preserve the nodes with highest $N_{eff}$ scores entries per layer.

We performed numerical experiments using a KAN network with the initial width $[28 \times 28, 64, 10]$ on the MNIST dataset. After training for 10 epoches the validation loss reached 0.0923 with a validation accuracy of 97.15%. By applying EMP to KAN, the network structure changed to $[28 \times 28, 47, 10]$ with the validation loss increasing to 0.1810 and accuracy dropping to 94.36%.

## 5.3 LLMs

In this subsection, following the experimental setup in Sun et al. (2024), we evaluate LLama (Touvron et al., 2023a) and LLama-2 (Touvron et al., 2023b) model families' perplexity (PPL) on Wikitext and zero-shot accuracy across 7 sub-tasks. We examine the models under the pruning criterion magnitude, Wanda and the corresponding criterion composed with EMP: EMP-magnitude and EMP-Wanda. We show the average sparsity for EMP methods, the average PPL change and the average accuracy change for all 7 models in Table 2. The detail sparsity and PPL for each model and the detail accuracy of each task is shown in Appendix B.

From Table 2, the EMP based methods are able to perform a comparable performance with the dense network. Notably, the EMP-magnitude method reduced the PPL and increased the accuracy compared with the fixed sparsity magnitude method, in the cost of lower sparsity.

## 5.4 FEATUREWISE EFFECTIVE PRUNING

In this section, we show that EMP can also be applied to features, by yielding a pruned feature with almost the same as the original one. We apply EMP to an RGB image by processing each channel (R, G, B) independently. For a given channel, we defined the score matrix $s \triangleq X_c - \mu_c$, where $X_c$ denotes the pixel values of channel $c \in \{R, G, B\}$, and $\mu_c$ is the corresponding channel mean. Two EMP-based pruning strategies were considered: EMP Global Magnitude and EMP Patch Magnitude.

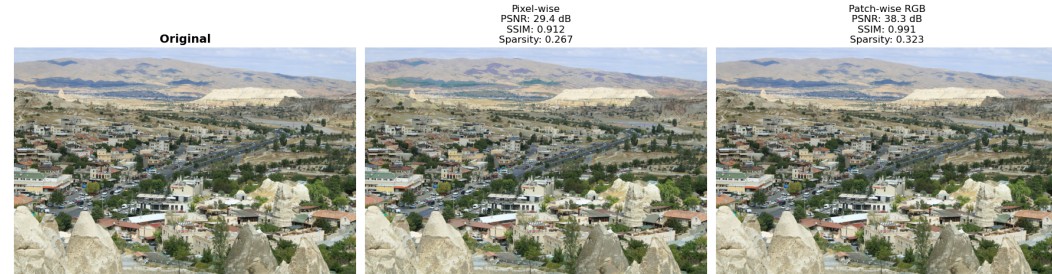

Figure 4: EMP magnitude pruning on an RGB image. Left: Original image (Figure Credit: https://www.pexels.com/photo/scenic-view-of-goreme-in-cappadocia-turkey-34012268/) Middle: EMP global magnitude pruning applied independently to each RGB channel. Right: patchwise EMP magnitiude pruning with local EMP applied on non-overlapping $4 \times 4$ patches. The global method retains $PSNR = 29.4dB$ and $SSIM = 0.912$ at sparsity $0.267$, while the patchwise method achieves higher fidelity ($PSNR = 38.3dB, SSIM = 0.991$) at increased sparsity $0.323$.

Table 2: $N_{eff}$ is an adaptive threshold for different criterion with different models. However, it yields near-constant sparsity within a method (std $\leq 0.33$). The EMP based methods, keeping the performance change small across methods, by utilizing the threshold $N_{eff}$, which reflects a different sparsity for different methods.

| Method | Avg Sparity(%) | Std | Avg $\Delta$PPL | Avg $\Delta$Acc.(%) |
|---|---|---|---|---|
| Wanda | 50.00 | 0.00 | +0.799 | -1.40 |
| Magnitude | 50.00 | 0.00 | +2.982 | -2.60 |
| EMP-Wanda | 40.47 | 0.33 | +0.678 | -1.50 |
| EMP-Magnitude | 36.63 | 0.04 | +0.752 | -0.93 |

EMP Global Magnitude uses the score matrix $s$ over the entire channel, and pruning was performed at the global scale, while EMP Patch Magnitude partitions the image into $4 \times 4$ non-overlapping patches, and EMP pruning was applied independently within each patch. After pruning, we restored the mean by adding $\mu_c$ back to each channel, followed by concatenation of the R, G, and B channels to reconstruct the pruned image.

To quantify the quality of the pruned image, we measured sparsity, structural similarity index (SSIM), and peak signal-to-noise ratio (PSNR) between the original and pruned images. The results are summarized in Fig 4.

## 6   CONCLUSION

In this paper, we developed a universal pruning threshold $N_{eff}$, which is agnostic to the scoring criterion, the network architecture, and the pruning paradigm. We show the theoretical tight lower bound for the preserved mass $s_{eff}$ through the simplex geometry. With the support of the lower bound of $s_{eff}$, we derived the upper bound for model loss change $\epsilon$ between the dense model and the EMP pruned model. In the experiments, we show that EMP will reach a $0$ loss change between the dense network and the pruned network, with different network architectures when using the magnitude criterion. By examining the coefficient $\beta$ with 6 numbers from $0.5$ to $2$, we verified the effective number $N_{eff}$ is the optimal setting. We show EMP can be paired with different pruning criterion, even feature level image pixels. In LLM, we examine the pruning performance of the LLama and LLama-2 model familes with the magnitude, wanda and the corresponding EMP criterions. The results indicate that EMP trades sparsity for pruned model performance, which is shown significantly in the comparison between the magnitude method and EMP-magnitude method.

## 7 REPRODUCIBILITY STATEMENT

All the experiments in this paper are reproducible. The code can be found in the supplementary materials and in this url: `https://anonymous.4open.science/r/Effective-model-pruning-F1C3`

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

## A  PROOF OF PROPOSITION 1

*Proof.* For $\nu = N$, the result is trivial because $A_\nu$ is a single point. Now assume $1 < \nu < N$. Let $\omega \in \tilde{\Delta}$ be fixed. Clearly, $p_\nu$ minimizes $\varphi_\nu$ along the interval $[\zeta_{[1]}, \zeta_{[N]}]$ (note that this interval is the longest edge of $\tilde{\Delta}$). Therefore, without loss of generality, we may assume $\omega \notin [\zeta_{[1]}, \zeta_{[N]}]$. Then, there exist indices $1 < i \leq \nu$ and $\nu < j \leq N$ such that $\omega_i > \omega_j$. Among the pairs $(i, j)$ satisfying this condition, select the one with smallest $j - i$ and smallest $j$. We donote this pair by $i = i(\omega)$ and $j = j(\omega)$. Let

$$H_\omega = \{x \in \mathbb{R}^N \colon \langle \zeta_{[N]} \mid x \rangle = 0, \langle \omega \mid x \rangle \geq 0\}.$$

It follows that $H_\omega$ is contained in the tangent space to $\tilde{\Delta}$, as well as

$$\|\omega + \epsilon x\| \geq \|\omega\| \tag{19}$$

for every $\epsilon > 0$.

We construct a vector $x \in H_\omega$ as

$$x_1 := 1, \ x_i := t, \ x_j := -(1 + t), \text{ and } x_k := 0 \text{ for } k \neq 1, i, j,$$

where

$$t = \frac{\omega_j - \omega_1}{\omega_i - \omega_j} \leq -1, \tag{20}$$

because $\omega_1 \geq \omega_i > \omega_j \geq \omega_N$. Also, note that $x$ is orthogonal to $\zeta_{[N]}$, implying that it is in the tangent space to the simplex $\Delta$. Furthermore, by the choice of the indices $i, j$, there exists $\epsilon > 0$ small enough that the ordering among the weights is preserved upon adding $\epsilon x$ to $\omega$, and therefore $\omega + \epsilon x \in \tilde{\Delta}$.

Consider the derivative of $\varphi_\nu$ at $\omega$ in the direction of $x$:

$$D\varphi_\nu\big|_\omega x = \nu \zeta_{[\nu]}^\top x = (1 + t) \leq 0.$$

The function $\varphi_\nu$ is linear. Therefore, setting $\epsilon_1 := \max\{\epsilon > 0 : \omega + \epsilon x \in \tilde{\Delta}\}$ gives rise to a point $\omega^1 := \omega + \epsilon_1 x$ with $\varphi_\nu(\omega^1) < \varphi_\nu(\omega)$ and with the property that either $\omega_1 \in [\zeta_{[1]}, \zeta_{[N]}]$ or $j(\omega^1) - i(\omega^1) < j(\omega) - i(\omega)$. Therefore, the process of augmenting $\omega$ into $\omega_1$ may be repeated at most $j - i$ times, generating a sequence of vectors $\omega^1, \ldots, \omega^k$ with $k \leq j - i$ and $\omega^k \in [\zeta_{[1]}, \zeta_{[N]}]$, and such that $\varphi_\nu(\omega) > \varphi_\nu(\omega^1) > \ldots > \varphi_\nu(\omega^k)$. Since $\omega^k \in [\zeta_{[1]}, \zeta_{[N]}]$, we finally have $\varphi_\nu(\omega^k) \geq \varphi_\nu(p_\nu)$ (with equality only if $\omega \neq p_\nu$ in the first place), as $\|\omega_k\| \geq \|p_\nu\|$.

Given there exist such a point $p_\nu$, let the vector $u$ represent the direction from $\zeta_N$ point to $\zeta_{[1]}$ as

$$u = \left[\frac{N-1}{N}, -\frac{1}{N}, \ldots, -\frac{1}{N}\right].$$

Then the coordinate of $p_\nu$ is

$$
\begin{aligned}
p_\nu &= c + r_{\nu+1}\frac{u}{\|u\|} \\
&= \begin{bmatrix} \frac{1}{N} + \frac{1}{N}\sqrt{\frac{(N-1)(N-\nu-1)}{\nu+1}} \\ \frac{1}{N}\left(1 - \sqrt{\frac{N-\nu-1}{(N-1)(\nu+1)}}\right) \\ \cdots \\ \frac{1}{N}\left(1 - \sqrt{\frac{N-\nu-1}{(N-1)(\nu+1)}}\right) \end{bmatrix}^\top
\end{aligned}
$$

and the one-norm of the projection $T_\nu(p_\nu)$ is

$$\varphi_\nu(\omega) = \|T_\nu(p_\nu)\|_1 = \frac{\nu}{N} + \frac{N-\nu}{N}\sqrt{\frac{N-\nu-1}{(\nu+1)(N-1)}}.$$

The proof is now complete. $\qquad \square$

Table 3: Network structure of FC models

| Model | Layer Width |
|-------|-------------|
| FC2   | $100, 10$ |
| FC5   | $1000, 600, 300, 100, 10$ |
| FC12  | $1000, 900, 800, 750, 700, 650, 600, 500, 400, 200, 100, 10$ |

## B  EXPERIMENT DETAILS

In this section, we provide the experimental details corresponding to Section 5. To further demonstrate that EMP is a context-agnostic pruning method, we additionally evaluate loss change and gradient saliency pruning mehtods on VGG16.

### B.1  TRAINING DETAILS

This section specifies the training configures of all the dense modeles used in this paper. We report dataset, architecture, optimizer, schedule, and pruning settings precisely.

**Datasets and preprocessing.**   We use the standard training and test splits of MNIST, FashionM-NIST, CIFAR-10, CIFAR-100, and TinyImageNet. Images are resized to the model's declared input size: 28 for MNIST/FashionMNIST, 32 for most CIFAR models, 224 for AlexNet (by upsampling), and 64 for TinyImageNet.

**Architectures.**   Fully connected baselines are FC2, FC5, and FC12 (Table 3). For CNNs, we use AlexNet and VGG16 on CIFAR-10. For CIFAR-100 and Tiny ImageNet we use ResNet-18/50.

**Optimization.**   Unless stated, batch size is 128 and epochs are 200. Optimizers and hyperparameters follow the configuration dictionary:

- **MNIST/FashionMNIST (FC2/5/12):** Adam, learning rate $10^{-4}$, no cosine schedule, no warmup, weight decay 0; FC2/5 trained for 5 epochs and FC12 for 10 epochs.
- **CIFAR-10:** FC5/FC12 with SGD, learning rate 0.01, no cosine schedule, no warmup, weight decay 0; VGG16 and AlexNet with SGD, learning rate 0.01, cosine schedule enabled, 5 warmup epochs, weight decay $5 \times 10^{-4}$.
- **CIFAR-100:** ResNet-18/50 with SGD, learning rate 0.1, cosine schedule, 5 warmup epochs, weight decay $5 \times 10^{-4}$.
- **Tiny ImageNet:** ResNet-18/50 with SGD, learning rate 0.01, cosine schedule, 5 warmup epochs, weight decay $5 \times 10^{-4}$.

SGD uses momentum 0.9. Adam uses its standard defaults unless otherwise noted.

**Learning-rate schedule.**   When cosine is enabled with warmup $W$, the step-$t$ learning rate for total $T$ epochs is

$$\text{lr}(t) = \begin{cases} \text{lr}_0 \cdot \frac{t}{W}, & 0 \le t < W, \\ \text{lr}_0 \cdot \frac{1}{2}\Big(1 + \cos\big(\pi \, \frac{t-W}{T-W}\big)\Big), & W \le t \le T. \end{cases}$$

Table 4 reports the detailed pruning results of EMP with $\beta = 1$, along with comparisons to their corresponding dense models.

### B.2  LLM

Meta LLama and LLama-2 checkpoints are from HuggingFace Hub. We load the pretrained, untuned weights and tokenizer; no fine-tuning or gradient updates occur. We run inference in $bfloat16$ with device map set to auto across $4\times$ NVIDIA $B200$ GPUs.

Table 4: Pruning results of FC models at $\beta = 1$ on MNIST and Fashion-MNIST.

| Dataset | Model | Dense Acc. (%) | Prune type | Acc. (%) | Sparsity (%) | $\Delta$ Acc. (%) |
|---------|-------|---------------|------------|----------|-------------|-------------------|
| MNIST | fc2 | 93.89 | block | 93.39 | 34.13 | -0.50 |
| | | 93.89 | global | 93.78 | 37.19 | -0.11 |
| | fc5 | 97.35 | block | 97.31 | 28.69 | -0.04 |
| | | 97.35 | global | 97.31 | 29.56 | -0.04 |
| | fc12 | 97.07 | block | 96.95 | 27.71 | -0.12 |
| | | 97.07 | global | 96.96 | 28.32 | -0.11 |
| F-MNIST | fc2 | 83.93 | block | 83.17 | 33.86 | -0.76 |
| | | 83.93 | global | 83.71 | 36.72 | -0.22 |
| | fc5 | 84.57 | block | 84.65 | 28.90 | +0.08 |
| | | 84.57 | global | 84.84 | 29.63 | +0.27 |
| | fc12 | 86.97 | block | 86.87 | 27.58 | -0.10 |
| | | 86.97 | global | 86.68 | 28.16 | -0.29 |

The detail of the sparsity, PPL, and mean accruacy across 7 subtasks are reported in Table 5. The detail accuracy for different tasks are reported in Table 6 for the LLama model family and Table 7 for the LLama-2 family models.

### B.3 ADDITIONAL EMP EXPERIMENTS

We evaluate the effectiveness of the $N_{eff}$ threshold on a VGG16 model trained on CIFAR-10. Three pruning criteria are considered: magnitude pruning, loss-change (Taylor) pruning, and gradient saliency pruning. We report accuracy, achieved sparsity, and FLOPs after pruning in Table 8.

For magnitude pruning $N_{eff}$ threshold yields a high sparsity regime with minimal accuracy drop, while for sensitivity-based criteria it trades a very low sparsity with the model performance.

Table 5: Detail perplexity and 7-task mean accuracy for LLama and LLama-2 families pruning model by Wanda, magnitude, EMP-Wanda, and EMP-magnitude.

| Model | Method | Sparsity (%) | PPL | ΔPPL | Mean Acc. (%) | ΔAcc (pp) |
|---|---|---|---|---|---|---|
| LLaMA 7B | Dense | 0.00 | 5.679 | +0.000 | 51.10 | +0.00 |
| | Wanda | 50.00 | 6.644 | +0.965 | 49.48 | -1.62 |
| | Magnitude | 50.00 | 11.002 | +5.323 | 48.42 | -2.68 |
| | EMP-Wanda | 40.60 | 6.362 | +0.683 | 50.34 | -0.76 |
| | EMP-Magnitude | 36.66 | 6.904 | +1.225 | 51.11 | +0.01 |
| LLaMA 13B | Dense | 0.00 | 5.090 | +0.000 | 53.60 | +0.00 |
| | Wanda | 50.00 | 5.836 | +0.746 | 52.00 | -1.60 |
| | Magnitude | 50.00 | 11.587 | +6.497 | 49.25 | -4.35 |
| | EMP-Wanda | 40.68 | 5.907 | +0.817 | 52.74 | -0.86 |
| | EMP-Magnitude | 36.58 | 6.666 | +1.576 | 52.02 | -1.58 |
| LLaMA 30B | Dense | 0.00 | 4.101 | +0.000 | 54.84 | +0.00 |
| | Wanda | 50.00 | 4.890 | +0.789 | 54.16 | -0.68 |
| | Magnitude | 50.00 | 5.553 | +1.452 | 53.57 | -1.27 |
| | EMP-Wanda | 40.18 | 4.687 | +0.586 | 52.90 | -1.94 |
| | EMP-Magnitude | 36.60 | 4.511 | +0.410 | 54.82 | -0.02 |
| LLaMA 65B | Dense | 0.00 | 3.531 | +0.000 | 59.28 | +0.00 |
| | Wanda | 50.00 | 4.267 | +0.736 | 57.40 | -1.88 |
| | Magnitude | 50.00 | 4.724 | +1.193 | 55.83 | -3.45 |
| | EMP-Wanda | 39.99 | 4.060 | +0.529 | 57.08 | -2.20 |
| | EMP-Magnitude | 36.61 | 3.865 | +0.334 | 56.95 | -2.33 |
| LLaMA-2 7B | Dense | 0.00 | 5.470 | +0.000 | 51.59 | +0.00 |
| | Wanda | 50.00 | 6.410 | +0.940 | 50.27 | -1.32 |
| | Magnitude | 50.00 | 9.712 | +4.242 | 48.52 | -3.07 |
| | EMP-Wanda | 41.07 | 6.513 | +1.043 | 49.78 | -1.81 |
| | EMP-Magnitude | 36.70 | 6.561 | +1.091 | 51.16 | -0.43 |
| LLaMA-2 13B | Dense | 0.00 | 4.881 | +0.000 | 53.64 | +0.00 |
| | Wanda | 50.00 | 5.591 | +0.710 | 52.12 | -1.52 |
| | Magnitude | 50.00 | 5.850 | +0.969 | 52.59 | -1.05 |
| | EMP-Wanda | 40.48 | 5.468 | +0.587 | 51.96 | -1.68 |
| | EMP-Magnitude | 36.62 | 5.162 | +0.281 | 52.93 | -0.71 |
| LLaMA-2 70B | Dense | 0.00 | 3.319 | +0.000 | 60.00 | +0.00 |
| | Wanda | 50.00 | 4.026 | +0.707 | 58.81 | -1.19 |
| | Magnitude | 50.00 | 4.514 | +1.195 | 57.70 | -2.30 |
| | EMP-Wanda | 40.32 | 3.821 | +0.502 | 58.74 | -1.26 |
| | EMP-Magnitude | 36.66 | 3.662 | +0.343 | 58.57 | -1.43 |

Table 6: Detailed zero-shot accuracy across 7 tasks (LLaMA).

| Model | Method | BoolQ | RTE | HellaSwag | WinoGrande | ARC-e | ARC-c | OBQA | Mean |
|---|---|---|---|---|---|---|---|---|---|
| LLaMA 7B | Dense | 69.63 | 53.07 | 54.25 | 49.17 | 67.05 | 38.74 | 25.80 | 51.10 |
| | Wanda | 70.24 | 52.71 | 51.12 | 50.67 | 62.25 | 35.75 | 23.60 | 49.48 |
| | Magnitude | 67.25 | 52.71 | 48.95 | 50.04 | 60.73 | 34.64 | 24.60 | 48.42 |
| | EMP-Wanda | 72.05 | 53.07 | 52.59 | 48.62 | 64.86 | 37.63 | 23.60 | 50.34 |
| | EMP-Magnitude | 72.81 | 53.43 | 52.94 | 49.41 | 65.32 | 37.46 | 26.40 | 51.11 |
| LLaMA 13B | Dense | 66.36 | 54.51 | 57.46 | 48.07 | 74.33 | 43.86 | 30.60 | 53.60 |
| | Wanda | 68.32 | 53.07 | 54.87 | 47.83 | 70.24 | 40.87 | 28.80 | 52.00 |
| | Magnitude | 65.57 | 51.99 | 49.47 | 49.64 | 62.54 | 36.52 | 29.00 | 49.25 |
| | EMP-Wanda | 68.20 | 57.04 | 55.93 | 47.75 | 69.28 | 41.38 | 29.60 | 52.74 |
| | EMP-Magnitude | 65.84 | 53.07 | 54.95 | 47.99 | 70.37 | 42.32 | 29.60 | 52.02 |
| LLaMA 30B | Dense | 66.91 | 53.79 | 60.75 | 49.64 | 75.08 | 46.93 | 30.80 | 54.84 |
| | Wanda | 68.99 | 54.51 | 58.89 | 49.96 | 72.85 | 44.28 | 29.60 | 54.16 |
| | Magnitude | 70.46 | 52.35 | 56.40 | 50.20 | 71.93 | 43.43 | 30.20 | 53.57 |
| | EMP-Wanda | 67.25 | 52.35 | 59.31 | 50.36 | 71.00 | 42.66 | 27.40 | 52.90 |
| | EMP-Magnitude | 67.68 | 53.43 | 59.85 | 50.28 | 74.58 | 45.90 | 32.00 | 54.82 |
| LLaMA 65B | Dense | 80.31 | 66.79 | 62.32 | 50.20 | 74.87 | 46.67 | 33.80 | 59.28 |
| | Wanda | 80.15 | 60.29 | 60.49 | 50.43 | 74.16 | 45.31 | 31.00 | 57.40 |
| | Magnitude | 81.31 | 52.71 | 59.89 | 50.91 | 71.93 | 44.28 | 29.80 | 55.83 |
| | EMP-Wanda | 79.88 | 61.73 | 60.46 | 50.20 | 72.39 | 45.90 | 29.00 | 57.08 |
| | EMP-Magnitude | 81.04 | 54.51 | 61.81 | 50.36 | 73.70 | 46.25 | 31.00 | 56.95 |

Table 7: Detailed zero-shot accuracy across 7 tasks (LLaMA-2).

| Model | Method | BoolQ | RTE | HellaSwag | WinoGrande | ARC-e | ARC-c | OBQA | Mean |
|---|---|---|---|---|---|---|---|---|---|
| LLaMA-2 7B | Dense | 66.57 | 52.71 | 54.56 | 50.43 | 69.23 | 39.85 | 27.80 | 51.59 |
| | Wanda | 72.39 | 52.71 | 51.12 | 49.57 | 65.74 | 36.18 | 24.20 | 50.27 |
| | Magnitude | 64.86 | 53.79 | 50.49 | 49.49 | 62.79 | 34.04 | 24.20 | 48.52 |
| | EMP-Wanda | 68.72 | 53.07 | 51.32 | 49.80 | 65.66 | 37.12 | 22.80 | 49.78 |
| | EMP-Magnitude | 69.79 | 53.07 | 54.35 | 48.86 | 68.98 | 38.65 | 24.40 | 51.16 |
| LLaMA-2 13B | Dense | 66.82 | 52.71 | 57.54 | 48.54 | 73.32 | 45.39 | 31.20 | 53.64 |
| | Wanda | 66.06 | 52.71 | 55.13 | 49.57 | 70.79 | 41.38 | 29.20 | 52.12 |
| | Magnitude | 67.31 | 52.71 | 55.92 | 50.20 | 70.29 | 41.13 | 30.60 | 52.59 |
| | EMP-Wanda | 63.55 | 52.71 | 56.05 | 49.49 | 70.41 | 42.49 | 29.00 | 51.96 |
| | EMP-Magnitude | 65.60 | 52.71 | 57.57 | 49.09 | 72.31 | 42.83 | 30.40 | 52.93 |
| LLaMA-2 70B | Dense | 77.55 | 64.98 | 62.81 | 51.30 | 77.44 | 50.34 | 35.60 | 60.00 |
| | Wanda | 81.38 | 60.65 | 61.14 | 50.99 | 75.63 | 48.29 | 33.60 | 58.81 |
| | Magnitude | 82.45 | 57.76 | 60.15 | 52.09 | 73.36 | 45.90 | 32.20 | 57.70 |
| | EMP-Wanda | 75.81 | 65.70 | 61.91 | 50.43 | 74.75 | 48.21 | 34.40 | 58.74 |
| | EMP-Magnitude | 79.42 | 56.32 | 62.17 | 51.30 | 77.23 | 48.55 | 35.00 | 58.57 |

Table 8: Comparison of pruning criteria on VGG16 (CIFAR-10). Dense baseline: 91.12% accuracy, 626M FLOPs. Neff threshold uses $\beta = 1.0$.

| Method | Acc.(%) | Sparsity(%) | FLOPs |
|---|---|---|---|
| Dense | 91.12 | 0.0 | 626M |
| EMP-Magnitude | 90.98 | 59.5 | 397M |
| EMP-Loss change | 91.12 | 2.1 | 623M |
| EMP-Saliency | 90.97 | 25.7 | 579M |

Table 9: Pruning results on VGG16 trained on CIFAR-10. EMP pruning using $\beta \in \{0.8, 1.0, 1.2\}$.

| Criterion | Pruning Scheme | Acc.(%) | Sparsity (%) | FLOPs |
|---|---|---|---|---|
| Magnitude | Original (50%) | 90.99 | 50.0 | 436.7M |
| | Original (70%) | 90.64 | 70.0 | 348.7M |
| | Original (90%) | 69.08 | 90.0 | 221.0M |
| | Original (95%) | 10.00 | 95.0 | 157.9M |
| | EMP ($\beta = 0.8$) | 90.80 | 67.6 | 360.4M |
| | EMP ($\beta = 1.0$) | 90.98 | 59.5 | 396.9M |
| | EMP ($\beta = 1.2$) | 91.09 | 51.4 | 431.1M |
| Loss Change (Taylor) | Original (50%) | 88.14 | 50.0 | 490.8M |
| | Original (70%) | 49.86 | 70.0 | 410.1M |
| | Original (90%) | 10.00 | 90.0 | 261.0M |
| | Original (95%) | 10.00 | 95.0 | 169.8M |
| | EMP ($\beta = 0.8$) | 91.12 | 2.1 | 623.1M |
| | EMP ($\beta = 1.0$) | 91.12 | 2.1 | 623.1M |
| | EMP ($\beta = 1.2$) | 91.14 | 2.3 | 622.8M |
| Gradient Saliency | Original (50%) | 88.08 | 50.0 | 514.1M |
| | Original (70%) | 45.19 | 70.0 | 449.6M |
| | Original (90%) | 10.00 | 90.0 | 306.4M |
| | Original (95%) | 10.00 | 95.0 | 189.9M |
| | EMP ($\beta = 0.8$) | 91.10 | 20.6 | 590.7M |
| | EMP ($\beta = 1.0$) | 90.97 | 25.7 | 578.8M |
| | EMP ($\beta = 1.2$) | 90.82 | 30.9 | 566.2M |

