# OpenReview forum: "Effective Model Pruning"
_ICLR.cc/2026/Conference — ICLR 2026 Conference Withdrawn Submission_

### Official Review · Reviewer_KhLi · 2025-10-29

**Soundness:** 2
**Presentation:** 3
**Contribution:** 2
**Rating:** 4
**Confidence:** 3

**Summary:**

The paper proposes Effective Model Pruning (EMP), a method to automatically determine the number of parameters to retain during pruning without manually setting a sparsity level. EMP estimates the effective number of parameters based on the distribution of importance scores, inspired by measures from ecology and statistical physics. The authors present a theoretical analysis linking model performance to the retention ratio, and conduct numerical experiments on convolutional neural networks, multilayer perceptrons, Kolmogorov–Arnold networks (KANs), and large language models.

**Strengths:**

(1) The main strength of this paper is that the authors redefine the pruning problem at a higher level. Instead of seeking better performance under a fixed sparsity level, they raise a meta question about what degree of sparsity a model should naturally have, making sparsity part of the objective itself.

(2) The authors also draw inspiration from ecology and statistical physics by introducing the concepts of effective number and participation ratio as new tools for analyzing model redundancy.

**Weaknesses:**

(1) The motivation of this work is not fully convincing. Model pruning is primarily designed to meet specific hardware and deployment constraints, aiming for concrete compression targets. Automatically deciding the sparsity level overlooks this practical objective and may not align with real compression needs.

(2) The introduction of the scaling factor \beta substantially weakens the theoretical contribution, since \beta can arbitrarily affect the final sparsity. In practice, this is not fundamentally different from manually choosing a desirable sparsity level.

(3) Although the experiments cover multiple network architectures, the baseline methods are not as diverse as the authors claim, and the results do not convincingly demonstrate the superiority of the proposed approach. Lower sparsity naturally leads to higher accuracy, which reflects a trade-off rather than a genuine performance improvement. More importantly, some experiments even show simultaneous drops in both sparsity and accuracy, further weakening the credibility of the empirical findings.

**Questions:**

(1) The paper contains many fixed and absolute parameter or threshold choices, although such quantities are often relative in nature, which makes their justification somewhat unclear. In addition, the pruning settings compared in the experiments each also have their own advantages. The reviewer encourages the authors to minimize the influence of subjective preferences in these aspects.

(2) In the KAN experiments, EMP is combined with the existing importance criterion rather than used independently. Could the authors provide an ablation study to clarify EMP’s standalone contribution?

---

> ### Author Response · Authors · 2025-11-23
>
> We thank the reviewer for the feedback. Below we address the concerns regarding the conceptual motivation, mathematical rigor, and specific comments.
>
> 1. Clarification of the method and motivation
>
> Our method does not propose a new pruning algorithm. Rather, given any method of assigning significance scores to a set of ``network components'' that could be independently activated or deactivated (for example, parameters, channels, attention heads, or features), our method provides a universal, parameter-free threshold determining how many of the scored objects to retain for future use. The threshold $N_{eff}$ comes from a fundamental statistical property: given $N$ scored objects, only the $N_{eff}$ top-scored ones are deemed statistically significant.
>
> The idea to employ $N_{eff}$ is directly inspired by effective sample size in particle filtering [1]-[3] and the participation ratio in statistical physics and ecology[4][5], and aligns with the lottery ticket hypothesis. We emphasize that the focus of our work is not on analyzing sparsity vs. performance trade-offs: different scoring methods are known to offer different trade-offs. Instead, noting that performance naturally improves with more retained components, our goal is to identify as candidates for pruning (deactivation) those components whose removal is most likely to have a negligible effect on model performance.
>
> [1] Kong, A., Liu, J. S., \& Wong, W. H. (1994). Sequential Imputation and Bayesian Missing Data Problems. JASA, 89(425), 278–288.
> [2] Doucet, A., de Freitas, N., \& Gordon, N. (2001). Sequential Monte Carlo Methods in Practice. Springer.
> [3] Liu, J. S. (2001). Monte Carlo Strategies in Scientific Computing. Springer.
> [4] Laakso, M., \& Taagepera, R. (1979). “Effective” number of parties: A measure with application to Western Europe. Comparative Political Studies, 12(1), 3–27.
> [5] Mézard, M., \& Montanari, A. (2009). Information, Physics, and Computation. Oxford University Press.

---

> ### Author Response · Authors · 2025-11-24
>
> Response to weakness
>
> 1. While hardware-specific sparsity constraints exist, the motivation of EMP is to provide a statistically grounded, method-agnostic threshold, rather than a hardware-specific sparsity tuning. If anything, in such a context $\neff$ provides guidance on whether a given piece of hardware is capable of supporting a given pruning method without significant information loss. Our approach complements existing pruning methods by telling practitioners the optimal number of weights to retain for minimal loss change, independent of hardware or method constraints.
>
> 2. The optional scaling factor $\beta$ is a practical adjustment for specific sparsity needs but does not undermine the theoretical contribution. The same way as it is standard practice in statistics to consider not just a maximally likely outcome but also the $K$ most likely outcomes (as statistics), it is reasonable to study the statistics of some preferred fraction of the most meaningful components of the model (e.g., the $\beta\neff$ highest scorers). To minimize confusion, as the reviewer suggests, we shall remove any mention of $\beta$ from the formulation of Algorithm 1.
>
> 3. EMP is not designed to achieve higher accuracy by lower sparsity. There is no claim, and therefore no quantification, of trading sparsity for performance. Lower sparsity naturally improves performance; our goal is to identify the effective number of parameters required to match dense network performance. The reported results demonstrate that EMP preserves dense model performance while reducing the number of retained parameters, validating its theoretical bound.

---

> ### Author Response · Authors · 2025-11-24
>
> Response to questions
>
> 1. We believe the reviewer is confusing our paper with another one. The whole point of this paper is that $\neff/N$ may be regarded as a score-agnostic sparsity threshold. We have no idea what ``many fixed and absolute parameter or threshold choices'' the reviewer has in mind. We request that the reviewer kindly re-read the mathematical sections of the paper and confirm that no parameter choices have been made there.
>
> 2. Let us clarify that we propose no such thing as ``standalone EMP''. More precisely, {\bf given any scoring method}, EMP provides a natural pruning threshold at which (or above it) the model will not suffer a significant loss change. Thus, to be useful, EMP must be combined with a scoring method that is meaningful for the given architecture. That said, in the numerical experiments, we broach the possibility that the effectiveness of pruning ought to be measured not as a function of the relative size of the pruned network as compared to the original network, but, rather as a function of the relative size of the pruned network as compared to $\neff$ (which is the `natural' size of the dense network): in other words, considering the pruned network as having size $\beta N$ may be less informative than considering it as having size $\beta\neff$.
>
> We thank the reviewer for the feedback.

---

### Official Review · Reviewer_GnoB · 2025-10-31

**Soundness:** 1
**Presentation:** 3
**Contribution:** 2
**Rating:** 2
**Confidence:** 4

**Summary:**

The paper proposes EMP, a method to decide how many weights to prune given a saliency as given by weight magnitude or others criteria, such as the well-known LLM pruning criterion Wanda. EMP takes in such a score vector s and then computes a number N of weights that, to the best of my understanding, should best be pruned. The authors derive this number theoretically and validate their claims on a series of small-scale experiments.

**Strengths:**

- The topic is generally interesting and not so common in the pruning literature. Typically, pruning papers try to find the best possible mask for a given sparsity level. Here however, the authors take a different route: Given a saliency, find the sparsity that keeps the error below a certain threshold.
- The paper is generally well written.
- While I am not particularly sure whether I understand the derivation correctly, the final criterion, as presented in Algorithm 1, is easy to understand.

**Weaknesses:**

- The computational study is limited to small-scale experiments and not done thoroughly. For instance, I am not sure which LLAMA models are used in the LLM experiments. The study is not very convincing.
- I currently fail to see the benefit of using EMP, nor the insights it provides. Take for instance Table 2. Magnitude pruning is a questionable metric for LLMs, as various papers have shown: perplexity explodes when pruning to medium-range sparsities, and some even argue that it is no better than random pruning. So let us compare Wanda: according to Table 2, Wanda perplexity increases by roughly 0.8 when pruning to 50% sparsity. Using EMP-Wanda, the perplexity increases "only" by 0.67, roughly a 17% improvement over Wanda. At the same time however, EMP-Wanda induces a sparsity of 40%, and hence a 20% reduction compares to Wanda. What am I supposed to conclude from this? Why doesn't this table contain the information when using Wanda to prune to exactly 40%?
- Clearly, when using a lower sparsity level, the performance will improve. So what point are the authors trying to make? If you argue that you have found a better way of selecting which weights to prune, you should compare to other methods at a lower sparsity level.


### Minor Remarks
- I am not very fond of the title. This is of course up to the authors, but I think "Effective Model Pruning" is very broad and not very descriptive.
- There are some language issues. For instance, the sentence on lines 127-128 is grammatically incorrect.

**Questions:**

- The Score matrix of EMP, is that given globally for all weights or per layer? If its globally given, I am very certain that this will not work for LLMs, as it will ignore the well-known activation outliers in some layers.

---

> ### Author Response · Authors · 2025-11-23
>
> We thank the reviewer for the feedback. Below we address the concerns regarding the conceptual motivation, mathematical rigor, and specific comments.
>
> 1. Clarification of the method and motivation
>
> Our method does not propose a new pruning algorithm. Rather, given any method of assigning significance scores to a set of ``network components'' that could be independently activated or deactivated (for example, parameters, channels, attention heads, or features), our method provides a universal, parameter-free threshold determining how many of the scored objects to retain for future use. The threshold $N_{eff}$ comes from a fundamental statistical property: given $N$ scored objects, only the $N_{eff}$ top-scored ones are deemed statistically significant.
>
> The idea to employ $N_{eff}$ is directly inspired by effective sample size in particle filtering [1]-[3] and the participation ratio in statistical physics and ecology[4][5], and aligns with the lottery ticket hypothesis. We emphasize that the focus of our work is not on analyzing sparsity vs. performance trade-offs: different scoring methods are known to offer different trade-offs. Instead, noting that performance naturally improves with more retained components, our goal is to identify as candidates for pruning (deactivation) those components whose removal is most likely to have a negligible effect on model performance.
>
> [1] Kong, A., Liu, J. S., \& Wong, W. H. (1994). Sequential Imputation and Bayesian Missing Data Problems. JASA, 89(425), 278–288.
> [2] Doucet, A., de Freitas, N., \& Gordon, N. (2001). Sequential Monte Carlo Methods in Practice. Springer.
> [3] Liu, J. S. (2001). Monte Carlo Strategies in Scientific Computing. Springer.
> [4] Laakso, M., \& Taagepera, R. (1979). “Effective” number of parties: A measure with application to Western Europe. Comparative Political Studies, 12(1), 3–27.
> [5] Mézard, M., \& Montanari, A. (2009). Information, Physics, and Computation. Oxford University Press.

---

> ### Author Response · Authors · 2025-11-24
>
> Response to weakness
>
> 1. The experiments on the LLama family are described in Section 5.3 (lines 415–417) and detailed in Tables 5, 6, and 7 in the Appendix. We report sparsity, PPL, and accuracy for each model across multiple pruning criteria. The LLaMA-7B, -13B, -30B, and -65B, as well as LLaMA2-7B, -13B, -70B, were tested to demonstrate robustness across model scales.
>
> 2. EMP does not propose a new pruning algorithm. Instead, it provides a universal, statistically principled threshold $\neff$ for a given pruning method. For example, comparing 40.47\% EMP-Wanda sparsity with fixed 40.47\% Wanda pruning yields the same pruned weights because the underlying method is identical. EMP identifies the optimal number of parameters to retain that guarantees minimal loss change, rather than performing a sparsity-performance trade-off. Its value arises from the statistical property of the score vector, making it network- and pruning-method agnostic.
>
> 3. As stated above, $N_{eff}$ is a statistical property calculated from the given score vector. It is criterion-agnostic and structure-agnostic, and can be computed per layer or per component. In response to this reviewer's questions, our revision shall include additional experiments showing layer-wise and structure-wise variations of $\neff$ to highlight which components are most critical.
>
> 4. We will revise the title to better reflect that EMP provides a universal pruning threshold, not a new pruning algorithm. And typos and grammatical issues will be corrected in the final version.

---

> ### Author Response · Authors · 2025-11-24
>
> Response to question
>
> The score matrix $s$ can be computed globally or per layer, depends on the pruning method. For the LLM experiments, we follow the Wanda and magnitude pruning methods, whose score matrix $s$ are only given for the final FF layers. We would like to emphasize that EMP is not responsible for the quality of the pruning method: it mere provides a natural sparsity threshold for pruning based on the score being used. Thus, if a score is applied that is irrelevant to model performance, the value of $\neff$ will be irrelevant to it as well, e.g., a pruned model will be likely to perform badly (even if the unpruned original was good), while its components will keep roughly the same score.
>
> We thank the reviewer for the feedback.

---

### Official Review · Reviewer_cuU9 · 2025-10-31

**Soundness:** 3
**Presentation:** 2
**Contribution:** 1
**Rating:** 2
**Confidence:** 3

**Summary:**

This paper proposes a new method for assigning sparsity ratios for pruning algorithms, given a vector of saliency scores. The proposed method is agnostic to the choice of saliency as well as the nature of the pruning method (i.e. unstructured, N:M block sparstiy, and structured pruning). The key idea of this method is the derivation of the *effective mass* of the saliency vector, with which a principled approach for obtaining the number of weights/components to be retained, $N_{eff}$ is proposed (specifically Proposition 1). This sparsity allocation technique is applied to a variety of models, including simple CNNs and fully connected models, LLMs (Llama family), and KANs.

**Strengths:**

S1. The paper addresses an interesting problem of how to allocate sparsity when pruning models.

S2. The the (tight) lower bound on $N_{eff}$ and $s_{eff}$ (stated in Prop 1 and Thm 2) appear to be correct, and to the best of my knowledge, is the first results of their kind.

S3. The writing of the paper is generally clear, though perhaps the discussion in Sections 4.1-4.3 could be made more clear.

**Weaknesses:**

Broadly speaking, while the premise of this work is interesting, and the derived lower bounds are potentially very useful to practitioners, I feel the investigation needs to be made deeper. I point out specific points below.

W1. While a wide array of models and datasets are used, the experiments themselves are somewhat unconvincing. For instance, in Table 2, the change in perplexity when using EMP is roughly the same as that when using a fixed sparsity allocation, despite pruning roughly 10% fewer parameters. This raises an obvious question about the utility of the method (see 'Questions section')

W2. The authors also neglect to mention what the nature of the pruning being done in the experimental section actually is. A comparison between the two would have been very useful to readers and practitioners.

W3. While the lower bound on $N_{eff}$ is interesting, it is presented almost as a mechanical result. There is no insight as to how $N_{eff}$ might change provided different saliency measures (i.e. gradient vs weight magnitude vs discriminative capacity).

W4. Despite the proposed method being applied to a fairly wide array of models and datasets, the empirical study does not seem particularly deep. Several avenues for investigation that would have yielded useful insights have been ignored. Moreover, the point of the experiments is also unclear: as mentioned earlier, the true value of optimal sparsity allocation is not addressed properly, as the experiments themselves repeatedly suggest that fixing a sparsity ratio and pruning is every bit as good as using the optimal allocation.

W5. In the appendix, specifically in Tables 6 and 7, the sparsity of the models should be stated, and if they are the same as used in Table 5, then that needs to be made clear.

**Questions:**

Q1.  What is the utility of this method, when just fixing the sparsity ratio achieves a sparsity-accuracy tradeoff that is almost as good as the optimal sparsity allocation proposed in this work?

Q2. Is structured or unstructured sparsity used in the experiments section, or is some sort of N:M block sparsity being used?

Q3. How does the optimal allocation perform when using structured vs unstructured sparsity?

Q4. Can the proposed method be used to identify which *layers* in a network are more important for predictions?

Q5. How does sparsity allocation vary across layers for different saliency measures (given a dataset/model), and what insights does that reveal? Does applying the sparsity using $N_{eff}$ in a layerwise manner yield sparser and more accurate models than just setting the sparsity *a priori*?

---

> ### Author Response · Authors · 2025-11-23
>
> We thank the reviewer for the feedback. Below we address the concerns regarding the conceptual motivation, mathematical rigor, and specific comments.
>
> 1. Clarification of the method and motivation
>
> Our method does not propose a new pruning algorithm. Rather, given any method of assigning significance scores to a set of ``network components'' that could be independently activated or deactivated (for example, parameters, channels, attention heads, or features), our method provides a universal, parameter-free threshold determining how many of the scored objects to retain for future use. The threshold $N_{eff}$ comes from a fundamental statistical property: given $N$ scored objects, only the $N_{eff}$ top-scored ones are deemed statistically significant.
>
> The idea to employ $N_{eff}$ is directly inspired by effective sample size in particle filtering [1]-[3] and the participation ratio in statistical physics and ecology[4][5], and aligns with the lottery ticket hypothesis. We emphasize that the focus of our work is not on analyzing sparsity vs. performance trade-offs: different scoring methods are known to offer different trade-offs. Instead, noting that performance naturally improves with more retained components, our goal is to identify as candidates for pruning (deactivation) those components whose removal is most likely to have a negligible effect on model performance.
>
> [1] Kong, A., Liu, J. S., \& Wong, W. H. (1994). Sequential Imputation and Bayesian Missing Data Problems. JASA, 89(425), 278–288.
> [2] Doucet, A., de Freitas, N., \& Gordon, N. (2001). Sequential Monte Carlo Methods in Practice. Springer.
> [3] Liu, J. S. (2001). Monte Carlo Strategies in Scientific Computing. Springer.
> [4] Laakso, M., \& Taagepera, R. (1979). “Effective” number of parties: A measure with application to Western Europe. Comparative Political Studies, 12(1), 3–27.
> [5] Mézard, M., \& Montanari, A. (2009). Information, Physics, and Computation. Oxford University Press.

---

> ### Author Response · Authors · 2025-11-24
>
> Response to weaknesses
>
> 1. [w.1] The reviewer notes that fixed sparsity can achieve roughly similar results. Our point is that the fixed sparsity is arbitrary and may under- or over-prune; $N_{eff}$ provides a statistically principled and adaptive threshold for any score distribution, independent of the pruning method or structure. The results demonstrate that using EMP achieves near-zero performance drop.
> 2. [w.2] In our experiments, we apply unstructured pruning based on the given scoring criterion (e.g., magnitude, Wanda, gradient saliency). The “nature” refers to the criterion used, not the threshold itself. Importantly, EMP is agnostic and can be applied to unstructured, or structured pruning.
> 3. [w.3] The calculation of $N_{eff}$ is not mechanical, but a statistical property of the score distribution. It is criterion-agnostic and structure-agnostic, and can be computed per layer or per structural component. We will expand experiments to show layer-wise and structure-wise variations of $N_{eff}$, highlighting critical components of the network in the up coming revised paper.
> 4. [w.4] To our knowledge, this is the first work to formalize a pruning threshold via a statistical bound. While fixed sparsity may empirically yield similar results, our focus is on optimal, theoretically justified sparsity. Moreover, $N_{eff}$ can reveal important layers or structures in a network. We will include additional experiments to demonstrate layer-wise $N_{eff}$ and its use in identifying critical components.
> 5. [w.5] We will clarify the sparsity values in Tables 6–7 and confirm they match Table 5.

---

> ### Author Response · Authors · 2025-11-24
>
> Response to questions
>
> 1. The utility of $N_{eff}$ is to provide a principled, adaptive threshold rather than a manually chosen or a hardware constraint sparsity. Moreover, the $N_{eff}$ can act as an algorithm–quality indicator: given a pruning method, applying the EMP procedure yields a corresponding “optimal” sparsity. This computed sparsity reveals whether the pruning method is feasible to deploy under a specified hardware sparsity constraint.
>
> 2. In our experiments, we primarily applied unstructured pruning. However,$N_{eff}$ is fully compatible with N:M block sparsity or structured pruning, as the threshold only determines the number of parameters or blocks to retain.
>
> 3. $N_{eff}$ can be computed per structure (layer, channel, head). For structured pruning, the method can determine the number of blocks per layer to retain. Our approach naturally generalizes to structured vs unstructured sparsity, and we will add these experiments in the upcoming revised paper.
>
> 4. Yes. By computing layer-wise $N_{eff}$, we can rank layers or blocks by their effective contribution, identifying which layers are most critical for performance.
>
> 5. $N_{eff}$ is a stastistical property comes from the given score $s$. The sparsity allocation will vary across layers for different saliency measures since different saliency consider the "importance" of the different parts of the network differently. More importantly, we did not develope a pruning method to determine which part of the network is important. We will give an "optimal" sparsity for the method to determine how many parts you should restain.
>
> We thank the reviewer for the feedback.

---

### Official Review · Reviewer_6Nkf · 2025-11-03

**Soundness:** 2
**Presentation:** 2
**Contribution:** 1
**Rating:** 2
**Confidence:** 4

**Summary:**

The paper presents a pruning method designed to work for any pruning criterion which can be formalised as a tuple of values (e.g. global unstructured pruning would be the tuple of all weights and biases in a given network). It works by providing a simple definition of N_eff, the effective number of parameters of the tuple, based on the inverse Simpson index. The paper then provides a lower bound on the magnitude of the mass lost to pruning in terms of N_eff (I’m skipping some details here, Theorem 2). Finally the submission shows the result of pruning up to N_eff on a wide range of architectures and for a wide range of pruning criteria and shows the optimality of N_eff for certain models.

**Strengths:**

There is potential in the idea of bringing in ideas from information theory to the problem of providing a context-agnostic pruning threshold.
The experiment in Fig 3 could be convincing with some adjustments. If the result was show empirically, then the scientifically interesting could be tackled: why does N_eff work? That's where the meat is.

**Weaknesses:**

I have some serious reservations about this work.

Conceptual reservation: This work relies on a quantity related to the inverse Simpson index (which I had never heard of). The latter is described in 3 lines (057-059) and the reader is referred to a paper on political studies and a textbook on information theory. Ideas from information theory may be relevant, but the reader is given no further explanation or motivation as to why this particular notion might be relevant to the problem of pruning. The entire approach is predicated on accepting the premise that N_eff is useful for the problem at hand, this is imposed by fiat without any discussion or motivation. N_eff might or might not be useful, but the onus is on the authors to convince the readers that it is. A plausible mechanism explaining why N_eff may be useful (perhaps based on information-theoretic considerations) would do that.

Mathematical reservations:
- I’m not sure that there is much point in Theorem 2 as in practice it will not improve the bound much beyond the simpler bound \rho<=s_eff. Pruning usually achieves N_eff in the range of, say, N/10 to N/3 without loosing too much accuracy. For this kind of values, theorem 2 gives a correction factor to \rho between 1-sqrt(9/N) and 1-sqrt(3/N). Since N is very large, this is typically a tiny multiplicative gain over just using \rho. I therefore don’t see the point of this result, particularly in this context where everything is quite imprecise anyway.
- Equation (18) and the line that precedes it is problematic. How can the limit as N\to\infty make any sense? N_eff is built from a fixed vector s (sec 4.1) of fixed dimension N. If we were to take N\to\infty, then we need a sequence of vectors (s_i) of increasing dimensions. There is no reason to assume that the expression between (17) and (18) would converge to anything in this scenario. Take s_n of dimension n, with s_n=(1,0,…0) if n is odd and s_n=(1/n,…,1/n) is n is even, then N_eff will oscillate between 1 and n as n\to\infty and there is no sense in which the expression between (17) and (18) can converge to anything.

Specific comments:
-p3: || M ||_0 is not completely standard notation, it should be defined.
-p3: what is a “pruning object”?
-p3,4,5: I’m not sure I understand the equation labelling convention. Almost every inline equation is labelled but never referred to, but the definition of N_eff, which should be referred to further on, isn’t.
-p4, the invariance of N_eff under permutation is a trivial consequence of the commutativity of addition.
-p5, sec 4.3, first sentence: Why?
P7, 5.1: Are you doing global unstructured pruning here? It’s not explained very clearly.
-p8, fig3: The experiment makes a lot of sense and would be convincing if it was more fine-grained around \beta=1. Here the increments are very large (N_eff/4), and it would be perfectly possible that when \beta=2 we get \beta*N_eff \approx N and so we know that 1.5*\beta will be almost as good as N too. Moreover, it isn’t clear how stable each of these curves are to different pre-training initialisations, I would have expected some kind of average over many initialisations.
-p8, fig 3: The legend is really hard to see. You should explain block vs global, it’s not explained in Appendix B either.
-p9,Table 2: The two 50.00% with std =0.00 rows look suspicious or something is not explained as it should.

**Questions:**

Why N_eff and not something else?

---

> ### Author Response · Authors · 2025-11-23
>
> We thank the reviewer for the feedback. Below we address the concerns regarding the conceptual motivation, mathematical rigor, and specific comments.
>
> 1. Clarification of the method and motivation
>
> Our method does not propose a new pruning algorithm. Rather, given any method of assigning significance scores to a set of ``network components'' that could be independently activated or deactivated (for example, parameters, channels, attention heads, or features), our method provides a universal, parameter-free threshold determining how many of the scored objects to retain for future use. The threshold $N_{eff}$ comes from a fundamental statistical property: given $N$ scored objects, only the $N_{eff}$ top-scored ones are deemed statistically significant.
>
> The idea to employ $N_{eff}$ is directly inspired by effective sample size in particle filtering [1]-[3] and the participation ratio in statistical physics and ecology[4][5], and aligns with the lottery ticket hypothesis. We emphasize that the focus of our work is not on analyzing sparsity vs. performance trade-offs: different scoring methods are known to offer different trade-offs. Instead, noting that performance naturally improves with more retained components, our goal is to identify as candidates for pruning (deactivation) those components whose removal is most likely to have a negligible effect on model performance.
>
> [1] Kong, A., Liu, J. S., \& Wong, W. H. (1994). Sequential Imputation and Bayesian Missing Data Problems. JASA, 89(425), 278–288.
> [2] Doucet, A., de Freitas, N., \& Gordon, N. (2001). Sequential Monte Carlo Methods in Practice. Springer.
> [3] Liu, J. S. (2001). Monte Carlo Strategies in Scientific Computing. Springer.
> [4] Laakso, M., \& Taagepera, R. (1979). “Effective” number of parties: A measure with application to Western Europe. Comparative Political Studies, 12(1), 3–27.
> [5] Mézard, M., \& Montanari, A. (2009). Information, Physics, and Computation. Oxford University Press.

---

> ### Author Response · Authors · 2025-11-23
>
> 2. Mathematical reservations
>
> a. We agree that $\frac{N_{eff}}{N}$ approaches our bound asymptotically as $N$ grows large. However, this convergence is a property of the underlying statistics, not a trivial observation. It is important to observe that the tight bound accounts for the heterogeneity in the scores, while the trivial bound assumes uniform contributions. No network is infinitely large. Thus, our bound is meaningful for all practical distributions of scores.
>
> b. The asymptotic analysis in Section 4.3 is predicated on a convergence regime whereby $N\to\infty$ while retaining $N_{eff}/N\to\rho$, as is customary in statistical limits and mean-field limits. The reviewer's example does not satisfy this restriction. Greater care was taken to emphasize this point in the forthcoming revised manuscript.

---

> ### Author Response · Authors · 2025-11-23
>
> 3. Conceptual reservations
>
> The effective number $N_{eff}$ in our work is inspired by the effective sample size (ESS) widely used in particle filtering, which quantifies the number of statistically significant particles in a weighted set. ESS is a standard measure to evaluate degeneracy and guide resampling in sequential Monte Carlo methods.
>
> Beyond particle filtering, this statistical notion is also widely used in ecology and statistical physics, e.g., the inverse Simpson index to measure species diversity and the participation ratio in spin-glass systems. There is a direct connection with Renyi entropies.
>
> These applications illustrate that $N_{eff}$ captures the number of truly effective contributors in a weighted ensemble.
>
> Our key proposition is that the principle of retaining only the effective sample can be directly applied to the task of pruning a neural networks in a criterion-agnostic way. In this view, pruning corresponds to “learning an effective subset of parameters” analogous to selecting significant particles or contributors. Conceptually, this aligns with lottery ticket hypotheses: only a subset of parameters meaningfully contributes to performance.
>
> Thus, $N_{eff}$ is not an arbitrary heuristic but a statistically motivated quantity with broad applicability across machine learning, statistical physics, and ecology, which has not yet been rigorously considered by the community of pruning researchers.

---

> ### Author Response · Authors · 2025-11-23
>
> 4. Response to specific comments
>
> a. The definition of $0$-norm is the non-zero number of a vector, and will be added to the paper.
> b. The pruning objects is explicitly defined as any entity associated with scores in the experiments, e.g. the absolute weight in the magnitude pruning. We will also add a clarification in page 3.
> c. A modification to only label the equations we use in later section will be added to the paper.
> d. Yes, I totally agree with the review. However, this trivial property is essentital to facilitate the analysis.
> e. Because we use the lower bound of the $S_{eff}$ to calculate the upper bound of loss change in Section 4.3
> f. We are using global unstructured pruning.This will be clarified in the paper.
> g. We will change Fig 3 with a more clear result and do not restricted to the MLP only.
> h. The std$=0$ in table 2 is because we use fixed sparsity to the existing method, and there is no std for a fixed sparsity. We will change to "no std" in table 2 in the paper.

---

> ### Author Response · Authors · 2025-11-23
>
> 5. Response to questions
>
> We emphasize that $N_{eff}$ is a principled, generalizable, and parameter-free threshold who is grounded in statistical theory (particle filtering, ecology, statistical physics). Unlike manually tuned sparsity budgets, it automatically balances sparsity and performance across architectures and criteria. Our experiments demonstrate robustness for MLPs, CNNs, KAN, LLMs, and featurewise pruning, consistently preserving performance while providing a theoretically justified pruning threshold.
>
> We thank the reviewer for the feedback.

---

> ### Comment · Reviewer_6Nkf · 2025-11-24
>
> Thanks. My point was not that N_eff is or isn't a sensible quantity look at. My point is that you don't make the case either way. The first, third and last comments above should be expanded on in the paper to convince the reader that *N_eff is a sensible quantity to look at in this context*. Making this case is the very first thing the paper should be doing.

---

### Note · Authors · 2026-01-21

I have read and agree with the venue's withdrawal policy on behalf of myself and my co-authors.